# Urban heat islands in China enhanced by haze pollution

Chang Cao[1,2], Xuhui Lee[1,2], Shoudong Liu[1], Natalie Schultz[2], Wei Xiao[1,2], Mi Zhang[1] & Lei Zhao[1,2,3]

The urban heat island (UHI), the phenomenon of higher temperatures in urban land than the surrounding rural land, is commonly attributed to changes in biophysical properties of the land surface associated with urbanization. Here we provide evidence for a long-held hypothesis that the biogeochemical effect of urban aerosol or haze pollution is also a contributor to the UHI. Our results are based on satellite observations and urban climate model calculations. We find that a significant factor controlling the nighttime surface UHI across China is the urban–rural difference in the haze pollution level. The average haze contribution to the nighttime surface UHI is $0.7 \pm 0.3$ K (mean $\pm 1$ s.e.) for semi-arid cities, which is stronger than that in the humid climate due to a stronger longwave radiative forcing of coarser aerosols. Mitigation of haze pollution therefore provides a co-benefit of reducing heat stress on urban residents.

[1] Yale-NUIST Center on Atmospheric Environment and Collaborative Innovation Center of Atmospheric Environment and Equipment Technology, Nanjing University of Information Science and Technology, Nanjing, Jiangsu 210044, China. [2] School of Forestry and Environmental Studies, Yale University, 195 Prospect Street, New Haven, Connecticut 06511, USA. [3] Woodrow Wilson School of Public and International Affairs, Princeton University, Princeton, New Jersey 08544, USA. Correspondence and requests for materials should be addressed to X.L. (email: xuhui.lee@yale.edu).

The urban heat island (UHI) represents one of the most pronounced surface climate changes caused by human activities[1]. The mechanism underlying the UHI formation is generally thought to be biophysical in nature, arising from large differences between rural and urban land in surface properties, including sensible heat dissipation or convection efficiency, evaporative cooling, sunlight reflection and artificial heating[2,3]. The main contributors to the daytime UHI are reductions in sensible heat convection efficiency and evaporative cooling in urban land[3–5]. At night, the heat released from energy use and from solar energy stored in buildings is a major biophysical factor responsible for urban warming. The UHI increases the number and the intensity of heat waves in cities, thus aggravating heat stress on urban residents[6].

Cities are also the dominant source of anthropogenic aerosols having large impacts on the biogeochemistry of the atmosphere[7,8]. The haze plume formed from urban aerosols alters regional precipitation patterns outside the city[9] and contributes to radiative forcing on the global climate[10]. The haze biogeochemical effect on the climate of urban land itself is, however, still not well understood, in large part because of the difficulty in disentangling the opposing effects aerosols have on the surface shortwave and longwave radiation[11]. Aerosols generally reduce the amount of shortwave radiation reaching the ground surface, creating a cooling effect on the surface. However, they are much more effective in absorbing and emitting radiation than water vapour and greenhouse gases in the longwave atmospheric window (wavelength range 8–12 μm) under specific conditions, thus having the potential to increase the longwave radiation energy received at the urban land surface[12]. The overall effect depends on the initial particle size and size growth due to ageing and absorption of water vapour[13].

So far, a quantitative evaluation of the haze contribution to the UHI has not been attempted. There are two difficulties. First, standard urban land surface models[14] do not include parameterizations for incoming shortwave and longwave radiation, because these are properties of the atmosphere aloft, not biophysical properties of the surface itself. Second, the aerosol radiative forcing is not a prognostic variable in atmospheric data assimilation models and in most climate models.

Cities in China are burdened with unprecedented levels of aerosol pollution. Here we present an empirical analysis using satellite observations to show that urban haze pollution is a contributor that intensifies the UHI in China at night. In the analysis, the surface UHI intensity $\Delta T$ is the difference in surface temperature between the urban and the adjacent rural land[3], and haze pollution is measured by the aerosol optical depth (AOD). We then use an urban climate model in conjunction with an observation minus reanalysis (OMR) method for aerosol longwave radiation enhancement, to quantify the haze contribution to the surface UHI intensity in three climate zones (humid, semi-humid and semi-arid) across China. We find that haze pollution intensifies the nighttime UHI in China through an increase in incoming longwave radiation. This warming effect is greater in semi-arid cities compared with cities in humid and semi-humid climates.

## Results

**Drivers of UHI spatial variations.** We analysed the annual mean $\Delta T$ measured by the Moderate Resolution Imaging Spectroradiometer (MODIS) instrument on board of the Aqua satellite from 2003 to 2013 for 39 cities across Mainland China (Fig. 1a). Being a proxy for heat release from anthropogenic sources and from solar energy stored in buildings, population is a predictor frequently used to explain city-to-city variations in the nighttime

UHI intensity observed by satellites[4,15–17] and by weather stations[2]. Our results indicate that, in contrast to studies reported for other regions of the world[4,15], city population is actually a poor predictor of the nighttime $\Delta T$ variations among these cities (Fig. 1b). Shanghai, located in southeast China and the largest city we analysed (population 14 million), shows a weak surface UHI (1.5 K), whereas Hami, a small city (population 0.45 million) in northern China, exhibits one of the strongest surface UHIs (5.0 K). The overall correlation between population and nighttime $\Delta T$ is not statistically significant ($P = 0.17$).

Another unusual feature is the diurnal variations of the surface UHI. The MODIS-derived nighttime $\Delta T$ (3.4 ± 0.2 K, mean ± 1 s.e.) is higher than the daytime value (2.1 ± 0.3 K; $P < 0.001$). The diurnal contrast is especially striking for cities in the semi-arid climate, where the mean nighttime $\Delta T$ is 4.0 ± 0.4 K, but the daytime $\Delta T$ is only 0.3 ± 0.5 K ($P < 0.001$; Fig. 2c,f). The diurnal patterns in China differ from those observed by satellites for North America[18], Europe[19,20], South America[20] and Oceania[20] where the daytime surface UHI is stronger than the nighttime UHI and where the UHI of semi-arid cities is generally weak at night[4,17,18]. Our results are broadly consistent with the UHI spatial pattern documented for China in a previous study using a shorter MODIS time series[21], although our UHI intensity is generally greater, because we only used pure urban and rural pixels to calculate $\Delta T$.

One explanation for these unique surface UHI patterns is related to haze pollution. We find that the spatial variations of the annual mean nighttime $\Delta T$ are significantly correlated with the difference in AOD between urban areas and the adjacent rural land (Fig. 1c; $\Delta$AOD, urban AOD minus rural AOD; $P < 0.01$). Significantly positive correlation is also found between the summer nighttime $\Delta T$ and $\Delta$AOD ($P < 0.01$). Cities having a thicker haze layer than the surrounding rural environment tend to display a stronger UHI. We use $\Delta$AOD, because $\Delta T$ is also a perturbation signal in reference to the rural background. The AOD itself is not a good predictor of the $\Delta T$ variations ($P = 0.48$). Only after controlling for $\Delta$AOD does $\Delta T$ show significant dependence on population ($P < 0.01$).

There is no evidence of haze enhancement on the daytime $\Delta T$. The correlation between the annual daytime $\Delta T$ and $\Delta$AOD is poor ($P = 0.43$; Fig. 3). Repeating the correlation analysis for the summer season reveals similarly poor correlation ($P = 0.50$). Instead, the most important factors explaining the daytime $\Delta T$ variations are population, urban–rural difference in normalized difference in vegetation index and cropland fraction of the rural background ($P$-values < 0.001). Annual mean precipitation exerts a strong control on the daytime $\Delta T$ in North America ($P < 0.001$; ref. 3) but a weak control in China ($P = 0.06$). This regional difference can be explained by the fact that cropland is a more prominent non-urban land cover in China than in North America. Irrigation is commonplace in China, with 48% of the farmland receiving water from irrigation in addition to water supplied by rain[22]. Domesticated plants supported by irrigation water do not behave in the same way as natural ecosystems in terms of surface energy exchanges. After excluding cities whose adjacent rural area consists of > 50% cropland pixels, precipitation becomes a significant controlling factor (linear correlation = 0.57, $P < 0.01$, number of observations = 21). The UHI dependence on precipitation and irrigation highlights the important role of the rural background environment in calculating $\Delta T$.

**Attribution of the haze effect.** Figure 1c can be viewed as empirical evidence supporting the long-held hypothesis that urban haze pollution is a contributor that intensifies the UHI[2]. We now make a quantitative attribution of the haze contribution to the nighttime $\Delta T$ by combining climate model calculations with

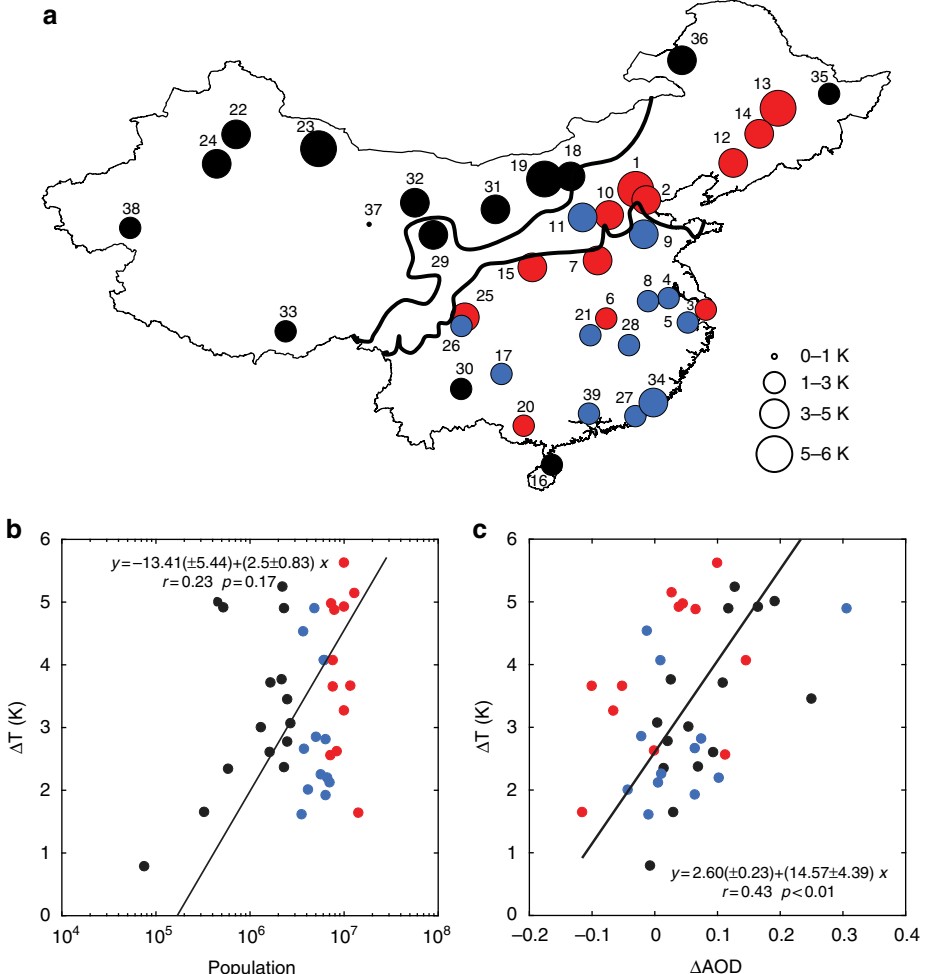

**Figure 1 | Nighttime MODIS surface urban heat island intensity in mainland China from 2003 to 2013.** (**a**) Spatial variation of the annual-mean nighttime MODIS-derived surface Urban Heat Island (UHI) across mainland China (K). (**b**) Surface UHI intensity relationship with population. (**c**) Surface UHI dependence on urban–rural AOD difference. Red, blue and black circles indicate large (population > 7 million), medium (3–7 million) and small cities (< 3 million), respectively. The two thick lines in **a** mark the boundary of three Köppen–Geiger climate zones (humid, semi-humid and semi-arid from south to north). Lines in **b**,**c** are linear regression with regression statistics noted. Errors on the regression parameters are 95% confidence bounds.

analysis of surface longwave radiation observations. The surface radiation data are used in conjunction with the surface longwave radiation calculated by an atmospheric data assimilation model, to obtain the sensitivity of $L_\downarrow$ to AOD, that is, the amount of enhancement in $L_\downarrow$ in response to a unit increase in AOD (Methods). Here, $L_\downarrow$ is the downward longwave radiation received by the surface including emissions and scattering of air molecules and aerosols. In the climate model, the urban land is parameterized as a separate land unit at the subgrid level. We force the model with an assimilated atmosphere and save the surface energy balance variables of urban and non-urban subgrid land units for offline UHI attribution[4]. The attribution method separates the contributions of external radiative forcing, energy redistribution via aerodynamic resistance-associated sensible heat convection and energy redistribution via evaporation[23].

In this framework, the aerosol effect is an external forcing similar to anthropogenic heat release and to changes in the surface shortwave radiation arising from the urban–rural surface albedo difference, and can be expressed as,

$$(\Delta T)_h = \frac{\lambda_0}{1+f} \Delta L_\downarrow \qquad (1)$$

where $(\Delta T)_h$ is haze contribution to the UHI intensity, $\lambda_0$ ($\approx 0.20\,\mathrm{K\,m^2\,W^{-1}}$) is the local intrinsic climate sensitivity,

$f$ is a dimensionless energy redistribution factor and $\Delta L_\downarrow$ is the urban–rural contrast in $L_\downarrow$ calculated as the product of the satellite-observed $\Delta$AOD and the longwave radiation sensitivity to AOD. Our AOD sensitivity values (Table 1) fall in the range of those calculated with radiative transfer models[24–27]. According to the observations of a ground-based aerosol remote-sensing network[28], the aerosol Ångström exponent is smaller in the semi-arid northwest Chinese cities, indicating larger particle sizes, than in cities in the humid central and eastern China. The sensitivity for the semi-arid climate zone is much higher than for the humid climate zone, confirming a stronger longwave radiative forcing of coarser particles[11,25]. Our estimates of $\Delta L_\downarrow$, ~1.1 and 8.0 W m$^{-2}$ for the cities in the humid and semi-arid climate zone, respectively, are lower than those reported from paired observations at urban and rural sites[29,30], because we did not consider the $L_\downarrow$ enhancement caused by a warmer urban boundary layer[31] and emissions from urban canopy walls[32]. In the model domain, $L_\downarrow$ represents the downward longwave radiation incident on a reference plane above the urban canopy, which is the first model grid height.

We estimate that the haze contribution to the nighttime $\Delta T$ is $0.70 \pm 0.26$ K (mean $\pm 1$ s.e.) for the semi-arid cities and is small for the other two groups of cities (Table 1). The larger $(\Delta T)_h$ in semi-arid climate is a result of less efficient energy redistribution

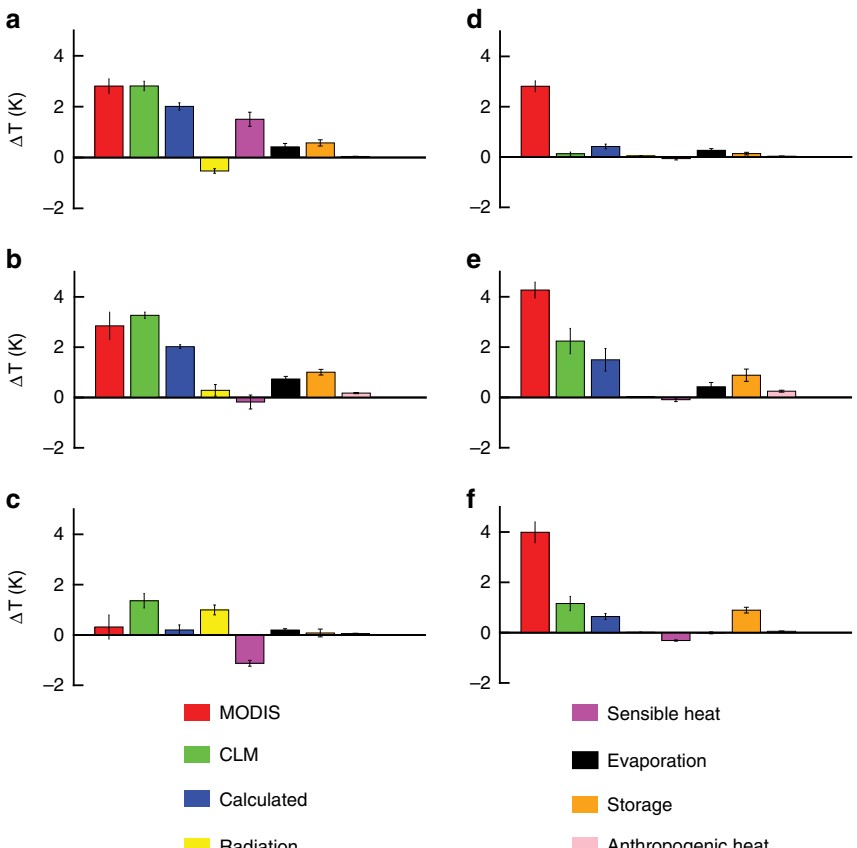

**Figure 2 | Urban Heat Island attribution and comparison with satellite observation.** Daytime Urban Heat Island (UHI) attribution in (**a**) humid region, (**b**) semi-humid region and (**c**) semi-arid region. Nighttime UHI attribution in (**d**) humid region, (**e**) semi-humid region and (**f**) semi-arid region. The red bar denotes the MODIS-derived UHI. The green bar denotes the UHI intensity determined online by the CLM model. The blue bar denotes the intensity calculated offline as the sum of the component contributions (changes in albedo as part of the radiation term (yellow bar), efficiency of sensible heat convection (magenta bar), evaporation (black bar), heat storage (orange bar) and anthropogenic heat release (pink bar) equation (2)) but excludes the haze contribution. Error bars are ±1 s.e.

(smaller $f$-values due to larger aerodynamic resistance and Bowen ratio; equation (5)) between the land and the atmosphere, a larger urban–rural contrast in pollution level and a stronger longwave radiative forcing of coarser aerosols.

In the semi-arid climate zone in China, both the urban and its adjacent rural area are affected by coarse mineral particles transported from the Taklimakan Desert and the Gobi Desert[33]. In the urban environment, road fugitive dust, construction-derived dust, and domestic heating and cooking are additional sources of coarse mode aerosols, explaining why the ratio of $PM_{10}$ to $PM_{2.5}$ concentration is greater in the semi-arid cities in Northwest China than in the humid cities in South China[34–36]. The urban–rural AOD difference observed here appears to result from these urban anthropogenic sources.

The haze contribution to the daytime $\Delta T$ is uncertain because of the opposing effects aerosols have on the surface shortwave and longwave radiation[11], but the lack of correlation between the daytime $\Delta T$ and $\Delta AOD$ (Fig. 3) suggests that it may be negligible. This inference is consistent with the model results. The daytime $\Delta T$ determined online by the Community Land Model (CLM) model, denoted as CLM in Fig. 2, is in good agreement with the MODIS-derived values (denoted as MODIS) for the cities in humid and semi-humid climates (Fig. 2a,b). For the cities in semi-arid climate, the model online result overestimates the observation, but the offline diagnostic calculation (denoted as Calculated), which is the sum of all the terms in equation (2), shows a good agreement (Fig. 2c). The overall agreement leaves

little room for an additional contribution due to the haze effect, implying that the relative reduction of the shortwave radiation in the city in reference to the rural background is roughly equal to the relative enhancement of the longwave radiation. This offsetting effect of aerosols on radiation has been reported previously by ref. 37 and the year-long observations at an urban–rural site pair support this interpretation[29]. Comparison of the model results and the MODIS observation for the summer also reveals very good agreement for the daytime.

According to the attribution diagnostics, the main contributor to the daytime UHI in the humid climate is the reduction in sensible heat convection efficiency of the urban land, not a reduction in evaporation. In the semi-arid climate, the role of convection is reversed, contributing to a cooling signal. These results are in agreement with those obtained previously for cities in North America[4].

In contrast to the daytime results, the modelled nighttime $\Delta T$ is too low in comparison with the MODIS observations (Fig. 2d–f), even though the same model is able to reproduce the observations in North America. The atmosphere in the model domain is free of haze pollution. With the inclusion of the haze contributions calculated offline (Table 1), the modeled $\Delta T$ is still biased low. One reason for the low bias is that the model scheme does not have a complete accounting of all sources of anthropogenic heat release in the urban land[38]. The anthropogenic heat flux is an important contributor to the nighttime UHI[21]. Another possibility is that equation (1) has omitted a dynamic mechanism

associated with the haze pollution. Aerosols are known to warm the atmosphere[12,39], potentially making the boundary layer air above the urban land more stable than that above the rural land and thus reducing the efficiency of energy dissipation from the urban surface to the atmosphere. The end result is an amplified UHI intensity. It appears that this stability mechanism is especially strong in the humid climate, as suggested by the large model bias error (Fig. 2d), although a definite answer will require an improved anthropogenic heat parameterization for China.

Our study implies that abatement of haze pollution has a co-benefit of reducing heat stress on urban residents. The UHI intensity is a perturbation signal in reference to the rural background temperature. A complete assessment of the haze effect must recognize that the rural atmosphere is also changing. The MODIS observation indicates that the rural AOD in China is 0.20 and 0.53, greater than that in North America in the semi-arid and the humid climate zones, respectively. The data in Table 1 suggest that the rural land in China may be receiving $\sim 15\,\mathrm{W\,m^{-2}}$ more longwave radiation energy at night than under haze-free conditions. Our subgrid energy balance analysis is not suited for quantifying the impact of the rural background change, because the change signal is regional in scale. The nighttime temperature in China increased at $0.47\,\mathrm{K}$ per decade from 1979 to 2012 (ref. 40), which is roughly twice the global

mean temperature trend for the same time period[41]. We hypothesize that haze pollution has contributed to the accelerated warming in China. If the hypothesis is proved valid, pollution abatement should also relieve heat stress on rural populations.

## Methods

**Satellite data.** We selected 39 cities in mainland China with the consideration that there is at least one city for each province (Supplementary Table 1). More cities were added for the three largest provinces (Xinjiang, Qinghai and Inner Mongolia), to ensure even distribution across the country. In our selection, cities located in mountainous regions were avoided. Of the selected cities, 19 are in humid climate, 9 in semi-humid climate and 11 in semi-arid climate, according to the Köppen–Geiger climate classification[42].

We used the (MODIS) Aqua land surface temperature (LST) product (MYD11A2), to retrieve paired urban and rural LST. The MYD11A2 product comprises 8-day clear-sky LST observations at 13:30 and 1:30 local time, with a 1 km spatial resolution. The study period is from 2003 to 2013. A set of $3 \times 3$ pixels was chosen from the urban core, except for three small cities for which we were only able to obtain one to three pure urban pixels. The surrounding rural areas were represented by up to four $3 \times 3$ pixel patches at the four sides of the city, but mountain and water pixels were avoided. The selected urban–rural pixel pairs do not differ by, on average, $>100\,\mathrm{m}$ in elevation and by $>0.1°$ in latitude.

Three features of the MYD11A2 product make it particularly suitable for the UHI detection. First, a MODIS cloud mask (MYD35) has been applied to filter out cloudy conditions; thus, cloud interferences are avoided. Second, a generalized split-window algorithm using two longwave bands in the atmospheric window is used to correct atmospheric water vapour and haze effects, and to reduce the sensitivity to errors in the surface emissivity[43]. The average bias (MODIS LST minus ground-based measurement of LST) is $-0.15 \pm 0.73\,\mathrm{K}$ in conditions of low haze pollution[44] (mean $\pm 1$ s.d., number of sites $n = 6$) and is not different statistically (two-sample $t$-test, $P = 0.98$) from the mean bias of $-0.17 \pm 1.66\,\mathrm{K}$ ($n = 4$) in conditions of high haze pollution[45]. Third, the brightness temperature has been corrected for surface emissivity to obtain the true LST. The surface emissivity data, developed by a MODIS science team[46], delineate the land surface into 17 categories (including a category for urban) and account for the bidirectional radiation distribution factor of each category. The impact of emissivity on the surface temperature retrieval has been discussed elsewhere[43,46].

Land cover change over time can complicate satellite UHI observations[47]. Steps were taken to ensure that the selected urban and rural pixels stayed as urban and rural throughout the duration of the study. The selected pixels were first validated by the MODIS land cover product (MCD12Q1, resolution 500 m) with the IGBP classification for year 2010 and then cross-checked against Google Earth. As no abandonment of urban land has occurred in these Chinese cities, these rural pixels should be rural in the years before 2010. The urban pixels were from the urban core; hence, they should be urban in the prior years as well as in the future years. To address the question of whether the pixels designated as rural in 2010 were converted to urban in the later years, we compared our selected rural pixels for each city against the MODIS classification for year 2013, the last year of our study period. Only the rural pixels for eight cities experienced some change from 2010 to 2013, but the change is small: $<5\%$ of the rural pixel selected based on the 2010 classification was converted to the urban class in 2013.

The MODIS Level 2 aerosol product (MYD04_3K) was used in this study. Compared with the nominal 10 km aerosol product, this product has a finer resolution of 3 km and is more suitable for determining urban and rural aerosol characteristics contrast. The AOD values were determined with the dark target algorithm[48]. Only high-quality (quality flag $= 3$) data were used. For each pair of city and its adjacent rural area, we calculated the averaged AOD from 2003 to 2013 and subtracted the rural mean AOD from that of the urban area to obtain $\Delta$AOD.

**The community land model and model simulations.** The model we used is the CLM (version 4.0) of the NCAR's Community Earth System Model (CESM) system[49]. Each gridcell in the CLM is configured with five land units: glacier, wetland, vegetated, lake and urban[50]. In this study, the urban and vegetated land units in the same gridcell were used to represent an urban–rural site pair. The model was driven by the atmospheric forcing data described in ref. 51. The simulation period

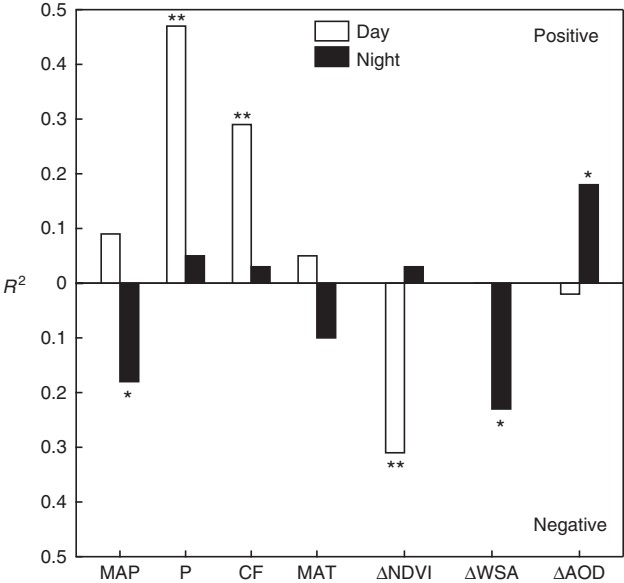

**Figure 3 | Variance of the annual mean daytime and nighttime $\Delta T$ explained by different biophysical drivers.** $\Delta$AOD, difference in aerosol optical depth; $\Delta$NDVI, urban–rural difference in normalized difference in vegetation index; P, population; $\Delta$WSA, difference in white sky albedo; CF, crop fraction; MAP, annual mean precipitation; MAT, annual mean air temperature. Positive correlations are shown in the upper panel and negative correlations in the lower panels. Confidence levels are denoted by $*P < 0.01$ and $**P < 0.001$.

**Table 1 | Estimate of the haze contribution to the night-time UHI for three climate zones.**

|  | Humid | Semi-humid | Semi-arid |
|---|---|---|---|
| AOD sensitivity (W m$^{-2}$ per unit AOD) | $31.9 \pm 3.0$ | $23.8 \pm 4.6$ | $61.8 \pm 4.9$ |
| $\Delta$AOD | $0.033 \pm 0.021$ | $0.033 \pm 0.020$ | $0.13 \pm 0.025$ |
| $f$ | $3.0 \pm 1.8$ | $1.7 \pm 0.9$ | $1.3 \pm 0.8$ |
| $(\Delta T)_{h}$ (K) | $0.05 \pm 0.12$ | $0.06 \pm 0.10$ | $0.70 \pm 0.26$ |

AOD, aerosol optical depth; UHI, urban heat island.
Error bounds are $\pm 1$ s.e.

began in 1972 and ended in 2004 after a 60-year spin-up. The model resolution (0.23° longitude × 0.31° latitude) was the finest allowed by this CESM version. Three small cities (Xining, Lhasa and Delhi) were ignored by the model, because their areal extent does not exceed 0.1% of the gridcell, a minimum area threshold to evoke the urban parameterization. To match the overpass time and the sky condition of the MODIS LST product, we only analysed model outputs at 1:00 and 13:00 local time, and under clear sky conditions (clearness index > 0.5).

**Offline attribution of UHIs.** We adopted the attribution method first developed for the investigation of local deforestation effects[23] and later extended to the study of the surface UHI of North American cities[4]. Briefly, the CLM subgrid model outputs were analysed in a surface energy balance framework that separates the contributions to the urban–rural temperature difference according to five biophysical drivers: net shortwave radiation or surface albedo change, change in sensible heat dissipation efficiency, evaporation reduction, heat storage change and anthropogenic heat release. Their contributions to $\Delta T$ are expressed as:

$$\Delta T = \frac{\lambda_0}{1+f}\Delta R_n^* + \frac{-\lambda_0}{(1+f)^2}(R_n^* - Q_s + Q_{AH})\Delta f_1 + \frac{-\lambda_0}{(1+f)^2}(R_n^* - Q_s + Q_{AH})\Delta f_2 + \frac{-\lambda_0}{1+f}\Delta Q_s + \frac{\lambda_0}{1+f}\Delta Q_{AH}$$

(2)

where $f$ is an energy redistribution factor, which is a function of Bowen ratio and the aerodynamic resistance (equation (3) below), $\lambda_0$ is a local climate sensitivity parameter ($\lambda_0 = 1/4\sigma T^3 \approx 0.2\,\mathrm{K\,W^{-1}\,m^2}$, where $\sigma$ is the Stephan–Boltzmann constant) and the terms on the right side of the equation represent contribution associated with difference in the net apparent radiation (term 1), sensible heat convection efficiency (term 2), evaporation (term 3), heat storage (term 4) and anthropogenic heat flux (term 5) between the urban and the rural area of the same grid. The change terms $\Delta f_1$ and $\Delta f_2$ are expressed as:

$$\Delta f_1 = \frac{-\lambda_0 \rho C_p}{r_a}\left(1 + \frac{1}{\beta}\right)\frac{\Delta r_a}{r_a}$$

(3)

$$\Delta f_2 = \frac{-\lambda_0 \rho C_p}{r_a}\frac{\Delta \beta}{\beta^2}$$

(4)

The surface UHI intensity was computed in two different ways. First, $\Delta T$ was calculated online by CLM as the surface temperature difference between the urban and the vegetated land unit in the same gridcell (the green bars in Fig. 2). In the second method, $\Delta T$ was computed as the sum of the individual contributions in the offline analysis according to equation (2) (the blue bars in Fig. 2).

The atmosphere in CESM is free of haze pollution, but the pollution effect can be quantified using the offline diagnostics according to equation (1). The most relevant diagnostic variable is the energy redistribution factor, $f$, which describes the efficiency of energy redistribution between the surface and the atmosphere, and is defined as,

$$f = \frac{\rho C_p}{4 r_a \sigma T_s^3}\left(1 + \frac{1}{\beta}\right)$$

(5)

where $\rho$ is air density, $C_p$ is specific heat of air at constant pressure, $r_a$ is aerodynamic resistance to sensible heat, $\sigma$ is the Stephan–Boltzmann constant, $T_s$ is the surface temperature and $\beta$ is Bowen ratio calculated as the ratio of the subgrid sensible heat to latent heat flux. These diagnostic variables are provided by the CLM model for every model grid. In a model grid where the $f$-value is higher, the surface UHI intensity will probably be lower, because energy is dissipated more efficiently from the surface to the atmosphere.

To estimate the haze enhancement on the surface UHI, we need to quantify the difference in the incoming longwave radiation between the urban and the rural area ($\Delta L_\downarrow$). We first determined the sensitivity of $L_\downarrow$ to AOD, that is, the amount of enhancement in $L_\downarrow$ in response to a unit increase in AOD, using the OMR method[52]. The reanalysis longwave radiation data were provided by the Modern-Era Retrospective Analysis for Research and Application. The observational data came from three ground sites in the ChinaFLUX network[53] representing the three climate zones (Yongfeng, 32.21° N, 118.67° E; Luancheng, 37.88° N, 114.68° E; Xilinguole 43.53° N, 116.66° E). Yongfeng (Jiangsu Province) and Luancheng (Hebei Province) are cropland sites and Xilinguole is a grassland site in Inner Mongolia. As Modern-Era Retrospective Analysis for Research and Application does not consider haze pollution but the observation was impacted by haze, some of the difference between the observed and reanalysed $L_\downarrow$ can be considered to result from haze pollution. In this study, the regression of the OMR $L_\downarrow$ against the MODIS AOD is given as [OMR $L_\downarrow$] = $a + b \times$ [AOD], where the intercept $a$ represents the reanalysis model bias[54] and the slope parameter $b$ was taken as the sensitivity of $L_\downarrow$ to AOD (Table 1). Next, the urban $L_\downarrow$ enhancement was calculated as the product of the AOD sensitivity and the observed AOD difference between the paired urban–rural pixels. Finally, the haze contribution to the UHI was calculated according to equation (1). The s.e. of $f$ and $\Delta L_\downarrow$ were calculated after averaging the values for cities belonging to the same climate zone. Uncertainty on $(\Delta T)_h$ was determined with a Monte Carlo procedure using 100,000 realizations and a Gaussian error distribution for all input variables.

**Data availability.** All the satellite data used in this study can be downloaded on the MODIS product website (https://lpdaac.usgs.gov/dataset_discovery/modis/modis_products_table/myd11a2). Other relevant data in this study are available from the authors.

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

## Acknowledgements

This research was supported by the National Science Foundation of China (grants 41475141 and 41575147), the Priority Academic Program Development of Jiangsu Higher Education Institutions (grant PAOD), the Ministry of Education of China (grant PCSIRT), a Visiting Fellowship from China Scholarship Council (to C.C.), a Yale University Graduate Fellowship (to N.S.) and a Princeton STEP Post-Doctoral Fellowship (to L.Z.). We acknowledge high-performance computing support from Yellowstone (ark:/85065/d7wd3xhc) provided by NCAR's Computational and Information Systems Laboratory, sponsored by the U.S. National Science Foundation.

## Author contributions

X.L. designed the research. L.Z. carried out the model simulation. C.C. performed the data analysis. S.L., N.S. and W.X. contributed ideas to the data analysis. M.Z. contributed the surface radiation data. C.C. and X.L. wrote the manuscript.

## Additional information

**Competing financial interests:** The authors declare no competing financial interests.

