## [Peer Review File · Nature Communications]

Reviewer Comments

Reviewer #1 (Remarks to the Author):

The presence of urban heat islands (UHI) is a well-known phenomenon, and the underlying mechanisms that lead to the phenomenon are myriads and complex. I applaud the authors' attempt to quantitatively untangle the driving forces with a special emphasis on the contribution from aerosols, a subject that is definitely of interest to the science community. Using MODIS and in situ observations and modeling approaches, the authors estimated the contributions of aerosols to UHI in 39 Chinese cities, providing numeric evidence of aerosols' role on UHI. However, the manuscript can be improved in the following areas.

(1) The manuscript is well-written, very succinct, and easy to follow with the caveat that the readers have to assume that everything they did is correct. Honestly, I find it very difficult to judge what they did is correct or not, given the succinctness of this manuscript. I think it is necessary for the authors to describe their method in detail so that other people can understand what they did and can repeat what did if so desired. I understand the journal might have a page limit for a paper, but the details should be supplied at least as supplemental materials.

(2) UHI has been studied extensively. For example, as early as 1998, Jacobson has studied the effects of aerosols on temperature profiles in Los Angeles using modeling approaches and found some interesting results on aerosols' role in regulating urban temperature. Some nice papers also appeared in the literature in recent years dealing with urbanization in China. Although none of the studies on Chinese cities was on partitioning the contributions of various driving forces to UHI per se, those studies do provide some references to the current research. I encourage the authors to reference these previous papers wherever appropriate. For example, it is not difficult to see the similarity and differences between this study and Zhou et al. (2014) on the spatial pattern and magnitude of the annual mean nighttime UHI.

Jacobson, M.Z., 1998. Studying the effects of aerosols on vertical photolysis rate coefficient and temperature profiles over an urban airshed. *Journal of Geophysical Research*

Zhou et al., 2014. Surface urban heat island in China's 32 major cities: spatial patterns and drivers. *Remote Sensing of Environment*

(3) The authors acknowledged that they found more pronounced UHI in China at night than that during the day and attribute this "unique pattern" to haze pollution and that their results are different from previous studies (e.g., Zhou et al., 2014). Why is there such a major difference? Please explain the reasons behind the difference and possible consequences on your conclusion as this is the foundation of your research.

(4) When using the annual mean, as done in this study, seasonal differences of aerosols might be obscured. Since aerosols have opposite effects on short- and long-wave radiation, is it possible that its net cooling effect in summer days and its net warming effect in winter days result in a negligible aerosol effect on annual daytime UHI after averaging? In other words, is it necessary to use sub-annual time intervals to study UHI?

(5) The effect of aerosols on the land surface energy budget is complex. Aerosols generally reduce solar radiation and enhance long-wave radiative forcing directly, while the net direct effect should vary with the types of aerosols. They can also directly and indirectly affect the formation and microphysical properties (e.g., cloud droplet size, cloud top albedo) of clouds, which may further

affect energy balance. Major uncertainties remain regarding the interactions between aerosols and clouds in general, and I wonder how these uncertainties are treated in the models and how they affect your conclusions.

(6) The definition of urban and its surrounding counterpart is the first step to accurately quantify UHI effects. What urban land cover (at what resolution, it does matter) and vegetation map did you use to define urban and rural area? Which years are these maps representing, are they consistent for these 39 cities selected? Data non-concurrency (i.e., the mismatch between land cover map and the LST map in time) may not be a big issue for natural ecosystems as their land surface alterations are generally not that rapid. However, it would be a huge issue for human-dominated ecosystems like urban system because land surface change in the urban environment (i.e., urban land expansion) is dramatic. For example, the urban pixels for a city you selected are based on 2010 urban map, and these pixels might not be urban (could be cropland which were converted to urban land in 2010) in 2003 or even in 2009. Such non-concurrency in data can lead to various biases in land surface conditions and UHI results (see Zhao et al., 2016). Did you consider the data concurrency issue such as the uncertainty in data concurrency on your results through Monte Carlo analysis?

Zhao et al., 2016. Data concurrency is required for estimation urban heat island intensity. Environmental Pollution.

Reviewer #2 (Remarks to the Author):

The authors use remotely sensed imagery to look at the area within and around urban areas in China to determine the size of the remotely sensed surface heat island. They use a numerical model to interpret the results.

A lot of key details are missing from the methods.

1) What processing has been done to the imagery with respect to atmospheric corrections

2) What corrections have been done with respect to the surface emissivity

Both of these issues need careful consideration with respect to the urban environment.

Given the source of aerosols could be either long range transport events (e.g. from the Gobi desert) or local sources (e.g. industry), there needs to be clear analysis that addresses these issues (both in terms of the data analysis but also in the preparation of the data used). This does not seem to be addressed. Analyses could consider upwind/downwind of the city with and without known haze events of different types. If the aerosols are from long range transport then there may not expect to have a large anthropogenic heat flux (AHF). However, if they are a local source then it is highly likely that there is also a large AHF. So again a more nuanced data analysis is needed.

Need to be clear what UHI is actually being referred to (see papers such as by Voogt on the number of types UHI and inconsistencies in terminology). This needs to be clear at all points in the paper (e.g. line 87?).

Figure captions are incomplete e.g. Figure 1 What period of data are analysed for this figure

Differences in the rural environment will also cause differences in the UHI calculated (given it's the difference between the two not the absolute nature of the urban environment)

L133 - What is the scale of this?

L143-145 - Rationale for parameters not clear

L149 - explain where Angstrom exponent is used

P7 where do your estimates represent vertically? Roof top? Ground levels? Higher?

L170-173 - Radiation does usually cancel - and has been found for a range of cities. this is what normally indicated in overviews of behaviour of radiative behaviour.

L179 - the change in convective behaviour should influence both turbulent heat fluxes (sensible and latent). Therefore, the moisture availability and the gradients must be critical. Rephrase to be clearer.

BL height change with density of aerosols

L222 - Reference

L224 - How processed?

L229 - How small? <3 million?

Emissivity assumptions; urban rural; atmospheric corrections?

L237 - reference

L248 - Relevance or implications of this period to the MODIS data

L273 - How defined?

L274 - daytime/nighttime?

L284 - so the model is perfect? Comment on size of uncertainties.

Figure 1 - Methods? Periods? What are data for?

Figure 2 - What's the sign convention that is being used? Is storage - storage heat flux. Why is radiation not included? By convection - I assume turbulent sensible heat flux is meant (evaporation or latent heat flux is a turbulent or convective heat flux also). The anthropogenic heat flux should be large in Chinese cities. What are the source of the aerosols? If regional then U-R difference does not hold. If Urban then AHF large.

Reviewer #3 (Remarks to the Author):

The paper "Urban heat islands in China enhanced by haze pollution" sets out to test the hypothesis that haze pollution, i.e., high concentrations of aerosols, in urban environments can contribute to the Urban Heat Island (UHI) effect by trapping some of the outgoing long-wave radiation in the urban domain. The authors claim to prove the existence of a significant impact of urban haze pollution (high aerosol concentrations) on the Urban Heat Island (UHI) effect by means of satellite data (MODIS). Furthermore, it aims to quantify the haze-pollution impact on UHI for a selected number of cities in China. Absorption of long-wave radiation depends to a large extent on the size of the particles which means that the retention of outward long-wave radiation in urban environments is due largely to the coarse fraction, i.e., dust particles and aged aerosol. Presumably, in coastal areas sea-salt aerosol could potentially contribute, too. The authors analyse satellite data to identify the haze-effect statistically in the observations. Quantification of the effect is attempted by applying a modelling approach in connection with re-analysis data and observations.

The paper presents novel research in the sense that it provides convincing evidence in support of a long-standing hypothesis. The work is novel also in that it extends existing research to a new domain - major cities in China - which due to the high level of aerosol pollution represent a promising research target for the considered effect. It is the relative dominance of haze or aerosol pollution in Asian cities, especially in China, that sets them apart from other major cities on the globe. In my opinion the paper is of moderate to significant interest to other researchers in the field. It presents convincing qualitative evidence for the existence of an impact of aerosol pollution on the UHI. That fact has important implications for pollution and urban climate change mitigation in that it outlines a co-benefit of reducing urban haze pollution.

In my opinion the claims made are convincingly underpinned by the evidence the authors have compiled in the paper, at least in a qualitative sense. The existence of a haze pollution impact on the UHI, at least during nighttime, is based on observational data and a simple but sound statistical analysis. The quantification also applies observations but does rely largely on a modelling approach and therefore has to be viewed, in my opinion, as a first estimate only which is exactly what the authors claim as well. Overall, I find the paper sufficiently convincing to justify publication in Nature Communications.

However, the main question that remains for me is in relation to the nature of the aerosol. The haze pollution effect on UHI is dominated by coarse aerosols, mainly dust I should think. It is vitally important to characterize and classify the sources of the coarse aerosol because the source determines the available mitigation options. Are the sources purely anthropogenic? If the main contributor is dust where does it originate? Is it road dust (e.g., unpaved road surfaces) or is coming from eroding soils outside the cities? If mitigation is a goal then these and further questions are essential. Maybe the authors could add a paragraph or two when they discuss the results.

The paper is presented in a clear fashion throughout. The abstract provides a concise and clear lead into the paper. With the exception of a few statements that could be improved and which I will discuss in detail under my specific comments the text is clear and easy to follow. In the introduction due attention is given to the existing literature with relevance to the subject and the background is discussed with sufficient detail. There is a clear thread throughout the section on the findings and the methodology is discussed in sufficient detail for the readers to understand and replicate the approach (provided they have access to the same data and model). I do not think that the paper should be shortened. In my opinion all the content is relevant and necessary. Figures and tables are very appropriate.

Therefore, I recommend for this paper to be accepted for publication in Nature Communications subject to a few revisions as outlined below.

Specific Comments:

I have already mentioned that it would be good to include a very brief discussion on the composition of the aerosol that make up the urban haze since it is vitally important for mitigation strategies. One or two paragraphs at max should be sufficient.

p3l56-59: I think this statement should be changed to "can be more effective under specific circumstances". I find it a bit of an overstatement or at least a very broad brush to claim that aerosols in general are more effective in absorbing/emitting long-wave radiation than greenhouse gases or water vapour. Also it should be explained here that the absorption effectiveness mostly depends on aerosol size: large aerosols are effective LW-absorbers. What about aged aerosols? How does water vapour uptake by aerosols change that property and what does this imply for the different climatic zones that have been investigated.

p6l111-113: I suggest to order these factors in order of decreasing magnitude with respect to R2: 1.) P, 2.) delta-NDVI, and 3.) CF.

p6l130: Most likely L-down is the downward long-wave radiation fraction as a result of long-wave radiation being absorbed and re-emitted in the downward direction by the aerosols. However, it is never really defined as such. Can it be assumed the reader knows? A brief definition would help in my opinion, if nothing else then just to remove any doubt.

p7l158: A bit more detail on the statistics would help. Mean plus standard error/deviation?

p8l169-171: It wasn't immediately obvious to me what the offline calculation was and how it is used in the context of the paper. If I understand correctly, the offline calculation is used to quantify the radiative effect which cannot be determined from the model simulation. The total radiation effect is then a combination of model and offline calculation. Maybe I was just not paying enough attention but a sentence or two of clarification would help.

p12l277-280: Could you add a discussion of other potential differences that may exist between

observations and the reanalysis data? How firmly is the assumption justified that the difference between model and reanalysis is ONLY due to aerosols? Surely, the model behind the reanalysis data has a different physical core than the urban model that is applied. There are other significant differences. A brief discussion of these differences and their potential impact on the findings would help I believe.

p15l351: Is there maybe another reference for this? I am not sure Discussion papers are acceptable references any longer.

Response to Referees

Response to Reviewer 1

The presence of urban heat islands (UHI) is a well-known phenomenon, and the underlying mechanisms that lead to the phenomenon are myriads and complex. I applaud the authors' attempt to quantitatively untangle the driving forces with a special emphasis on the contribution from aerosols, a subject that is definitely of interest to the science community. Using MODIS and in situ observations and modeling approaches, the authors estimated the contributions of aerosols to UHI in 39 Chinese cities, providing numeric evidence of aerosols' role on UHI. However, the manuscript can be improved in the following areas.

(1) The manuscript is well-written, very succinct, and easy to follow with the caveat that the readers have to assume that everything they did is correct. Honestly, I find it very difficult to judge what they did is correct or not, given the succinctness of this manuscript. I think it is necessary for the authors to describe their method in detail so that other people can understand what they did and can repeat what did if so desired. I understand the journal might have a page limit for a paper, but the details should be supplied at least as supplemental materials.

Thank you. In response, we have added a paragraph on MODIS data quality issues (**Lines 262-275**) and a paragraph on data consistency (**Lines 277-290**), and have expanded the description of the climate model (**Lines 332-346**).

(2) UHI has been studied extensively. For example, as early as 1998, Jacobson has studied the effects of aerosols on temperature profiles in Los Angeles using modeling approaches and found some interesting results on aerosols' role in regulating urban temperature. Some nice papers also appeared in the literature in recent years dealing with urbanization in China. Although none of the studies on Chinese cities was on partitioning the contributions of various driving forces to UHI per se, those studies do provide some references to the current research. I encourage the authors to reference these previous papers wherever appropriate. For example, it is not difficult to see the similarity and differences between this study and Zhou et al. (2014) on the spatial pattern and magnitude of the annual mean nighttime UHI.

Jacobson, M.Z., 1998. Studying the effects of aerosols on vertical photolysis rate coefficient and temperature profiles over an urban airshed. Journal of Geophysical Research

Zhou et al., 2014. Surface urban heat island in China's 32 major cities: spatial patterns and drivers. Remote Sensing of Environment

These two papers are indeed relevant to our study. They are cited as refs 12 and 21. The study by Jacobson supports our conclusion that longwave radiation associated with air pollutants enhances surface warming at night and can offset cooling due to the reduction in shortwave radiation during the day.

(3) The authors acknowledged that they found more pronounced UHI in China at night than that during the day and attribute this “unique pattern” to haze pollution and that their results are different from previous studies (e.g., Zhou et al., 2014). Why is there such a major difference? Please explain the reasons behind the difference and possible consequences on your conclusion as this is the foundation of your research.

We noted in our original submission that the diurnal UHI pattern in our study is different from those observed for other parts of the world (**Lines 99-102**). But our results are actually in broad agreement with the study of Zhou et al. (2014) for China. For example, we also found that the UHI intensity is correlated with precipitation and with the urban-rural white sky albedo difference (Figure 3). In their study, the UHI intensity of the semi-arid cities is much greater at night than during the day, again in agreement with our results (Figure 1a). However, our UHI intensity is generally larger than theirs because we only used pure urban and rural pixels to calculate ΔT whereas their selection was based on a somewhat arbitrary build-up intensity threshold and as such many of their urban and rural pixels were mixed pixels. In addition, we ensured that the two groups of pixels were matched in altitude and latitude (**Lines 259-260**), but no such screenings were used by Zhou et al.

We have added the following text when citing this study:

“Our results are broadly consistent with the UHI spatial pattern documented for China in a previous study using a shorter MODIS time series²¹, although our UHI intensity is generally greater because we used only pure urban and rural pixels to calculate ΔT .” (**Lines 102-105**)

(4) When using the annual mean, as done in this study, seasonal differences of aerosols might be obscured. Since aerosols have opposite effects on short- and long-wave radiation, is it possible that its net cooling effect in summer days and its net warming effect in winter days result in a negligible aerosol effect on annual daytime UHI after averaging? In other words, is it necessary to use sub-annual time intervals to study UHI?

In response to this question, we have repeated the analysis for the summer season when heat stress is expected. The UHI model result is similar to that shown in Figure 2. The correlation between UHI and ΔAOD is poor for the daytime ($r = -0.11$, $p = 0.50$) and statistically significant for the nighttime ($r = 0.49$, $p < 0.01$), which is consistent with the annual result.

We have added the following text:

- “Significantly positive correlation is also found between the summer nighttime ΔT and ΔAOD ($p < 0.01$).” (**Lines 110-111**)
- “The correlation between the annual daytime ΔT and ΔAOD is poor ($p = 0.43$; Figure 3). Repeating the correlation analysis for the summer season reveals similarly poor correlation ($p = 0.50$).” (**Lines 118-121**)
- “Comparison of the model result and the MODIS observation for the summer also reveals very good agreement for the daytime.” (**Lines 201-203**)

(5) The effect of aerosols on the land surface energy budget is complex. Aerosols generally reduce solar radiation and enhance long-wave radiative forcing directly, while the net direct effect should vary with the types of aerosols. They can also directly and indirectly affect the formation and microphysical properties (e.g., cloud droplet size, cloud top albedo) of clouds, which may further affect energy balance. Major uncertainties remain regarding the interactions between aerosols and clouds in general, and I wonder how these uncertainties are treated in the models and how they affect your conclusions.

We agree that the interaction between aerosols and the surface energy balance is complex. Our study shows that despite the complexity, the longwave radiation enhancement associated with coarse particles produces a discernable UHI signal. The MODIS observation and the model result are for clear sky conditions (Line 263-264, 310-313), so the cloud problem is avoided.

(6) The definition of urban and its surrounding counterpart is the first step to accurately quantify UHI effects. What urban land cover (at what resolution, it does matter) and vegetation map did you use to define urban and rural area? Which years are these maps representing, are they consistent for these 39 cities selected? Data non-concurrency (i.e., the mismatch between land cover map and the LST map in time) may not be a big issue for natural ecosystems as their land surface alterations are generally not that rapid. However, it would be a huge issue for human-dominated ecosystems like urban system because land surface change in the urban environment (i.e., urban land expansion) is dramatic. For example, the urban pixels for a city you selected are based on 2010 urban map, and these pixels might not be urban (could be cropland which were converted to urban land in 2010) in 2003 or even in 2009. Such non-concurrency in data can lead to various biases in land surface conditions and UHI results (see Zhao et al., 2016). Did you consider the data concurrency issue such as the uncertainty in data concurrency on your results through Monte Carlo analysis?

Zhao et al., 2016. Data concurrency is required for estimation urban heat island intensity. Environmental Pollution.

This is an excellent point. This paper has been added as ref 46. To address these issues, we have added a paragraph in the Methods section:

“Land cover change over time can complicate satellite UHI observations⁴⁶. Steps were taken to ensure that the selected urban and rural pixels stayed as urban and rural throughout the duration of the study. The selected pixels were first validated by the MODIS land cover product (MCD12Q1, resolution 500 m) with the IGBP classification for year 2010 and then cross-checked against Google Earth. Since no abandonment of urban land has occurred in these Chinese cities, these rural pixels should be rural in the years prior to 2010. The urban pixels were from the urban core, so they should be urban in the prior years as well as in the future years. To address the question of whether the pixels designated as rural in 2010 were converted to urban in the later years, we compared our selected rural pixels for each city against the MODIS classification for year 2013, the last year of our study period. Only the rural pixels for eight cities experienced some change from 2010 to 2013, but the change is small: less than 5% of the rural pixel selected based on the 2010 classification was converted to the urban class in 2013.” (Lines 277-290)

Response to Reviewer 2

The authors use remotely sensed imagery to look at the area within and around urban areas in China to determine the size of the remotely sensed surface heat island. They use a numerical model to interpret the results.

A lot of key details are missing from the methods.

(1) What processing has been done to the imagery with respect to atmospheric corrections

In response, we have added the following paragraph to the Methods section:

“Three features of the MYD11A2 product make it particularly suitable for the UHI detection. First, a MODIS cloud mask (MYD35) has been applied to filter out cloudy conditions, so cloud interferences are avoided. Second, a generalized split-window algorithm utilizing two longwave bands in the

atmospheric window is used to correct atmospheric water vapor and haze effects and to reduce the sensitivity to errors in the surface emissivity⁴². The average bias (MODIS LST minus ground-based measurement of LST) is -0.15 ± 0.73 K in conditions of low haze pollution (ref⁴³, mean ± 1 standard deviation, number of sites $n = 6$) and is not different statistically (two-sample t-test, $p = 0.98$) from the mean bias of -0.17 ± 1.66 K ($n = 4$) obtained in conditions of high haze pollution⁴⁴. Third, the brightness temperature has been corrected for the surface emissivity to obtain the true LST. The surface emissivity data, developed by a MODIS science team⁴⁵, delineate the land surface into 17 categories (including a category for urban) and account for the bidirectional radiation distribution factor of each category.” (Lines 262-275)

That the split window technique is successful at removing haze effects is further supported by our own analysis. At the Duolun FLUXNET site in the semiarid climate zone, the MODIS LST ($T_{S, \text{MODIS}}$) was in excellent agreement with the ground-based LST [$T_{S, \text{Ground}}$; $T_{S, \text{Ground}} = 0.98 (\pm 0.07) T_{S, \text{MODIS}} + 4.68 (\pm 20.86)$], $n = 26$, $r = 0.99$, $p < 0.001$]. The bias error was, however, independent of haze pollution [$T_{S, \text{MODIS}} - T_{S, \text{Ground}} = -4.40 (\pm 8.18)$ AOD $-1.23 (\pm 1.72)$, $r = -0.24$, $p = 0.28$].

(2) What corrections have been done with respect to the surface emissivity

Please see point 1 above.

(3) Analyses could consider upwind/downwind of the city with and without known haze events of different types. If the aerosols are from long range transport then there may not expect to have a large a large anthropogenic heat flux (AHF). However, if they are a local source then it is highly likely that there is also a large AHF. So again a more nuanced data analysis is needed.

The satellite data are not granulated enough for us to distinguish between wind directions and between different types of haze. So in this sense our study is one of climatology. To ensure that we were not biased against any particular wind direction, we selected rural pixels in all the four quadrants (north, south, east and west) outside the urban core as long as these pixels satisfy our latitude and altitude matching criteria.

It is true that different haze events may be associated with different AHF, but the satellite AOD data do not have information on aerosol type for us to establish a firm association. Instead, in our modeling framework, we treat the haze-enhanced surface long wave radiation separately from the AHF term in the surface energy balance. We do admit that ‘one reason for the low bias is that the model scheme does not have a complete accounting of all sources of anthropogenic heat release in the urban land³⁷.’ (Lines 215-217)

Please also refer to our response to Review 3, point 1 regarding haze type.

(4) Need to be clear what UHI is actually being referred to (see papers such as by Voogt on the number of types UHI and inconsistencies in terminology). This needs to be clear at all points in the paper (e.g. line 87?).

We now use the term “Surface UHI” throughout the paper. The Voogt paper is cited as ref 3. A precise definition is added: “In the analysis, the surface UHI intensity ΔT is the difference in surface temperature between the urban and the adjacent rural land³...” (Lines 73-75)

(5) Figure captions are incomplete e.g. Fig 1 What period of data are analysed for this figure

The study period is added to the figure caption.

(6) Differences in the rural environment will also cause differences in the UHI calculated (given it's the difference between the two not the absolute nature of the urban environment)

We agree completely with the reviewer on this point. This explains why two cities built identically but in two different climate zones have different UHI intensities (ref 4) and why the UHI is shown to depend on NDVI, local precipitation and irrigation status (Figure 3).

We have added a sentence to emphasize the reviewer's point:

"The UHI dependence on precipitation and irrigation highlights the important role of the rural background environment in calculating ΔT ." (Lines 133-134)

(7) L133 - What is the scale of this?

The resolution of the model is 0.23° longitude \times 0.31° latitude and only when the urban fraction bigger than 0.1% is the urban parameterization evoked. We mentioned this in the paper : "Three small cities (Xining, Lhasa and Delhi) were ignored by the model because their areal extent does not exceed 0.1% of the gridcell, a minimum area threshold to evoke the urban parameterization." (Lines 308-310)

(8) L143-145 - Rationale for parameters not clear

In response, we have enhanced the description of the modeling methodology by reproducing two key equations from refs 4 and 23 and by clarifying how the surface UHI was computed in the model domain. (Lines 333-337)

(9) L149 - explain where Angstrom exponent is used

The Ångström exponent, an optical parameter obtained from remote sensing, is a proxy parameter for aerosol particle size. In ref 28, the Angstrom exponent was analyzed: 'According to the observations of a ground-based aerosol remote sensing network²⁸, the aerosol annual Ångström exponent is smaller in the semiarid northwest Chinese cities, indicating larger particle sizes, than in the humid central and eastern China.' (Lines 160-163)

(10) P7 where do your estimates represent vertically? Roof top? Ground levels? Higher?

We have added one sentence to clarify this point: "In the model domain, L_{-} represents the downward longwave radiation incident on a reference plane above the urban canopy." (Lines 169-171)

(11) L170-173 - Radiation does usually cancel - and has been found for a range of cities. this is what normally indicated in overviews of behaviour of radiative behaviour.

By "cancellation" we mean that the relative change of the shortwave radiation in the city in reference to the rural background is roughly equal to the relative change in the incoming longwave radiation. We have modified the text here:

"The overall agreement leaves little room for an additional contribution due to the haze effect, implying that the relative reduction of the shortwave radiation in the city in reference to the rural background is roughly equal to the relative enhancement of the longwave radiation. The year-long

observations at an urban-rural site pair support this interpretation²⁹.” (Lines 197-201)

(12) L179 - *the change in convective behaviour should influence both turbulent heat fluxes (sensible and latent). Therefore, the moisture availability and the gradients must be critical. Rephrase to be clearer.*

Consistent with our earlier studies (refs 4 and 23), we used the term “convection” to mean turbulent transport of sensible heat. To avoid ambiguity, we have changed the term to “sensible heat convection efficiency”. (Line 206)

(13) L222 - *Reference*

We have added ref 41.

(14) L224 - *How processed?*

Please refer to point 1 above.

(15) L229 - *How small? <3 million?*

Correct.

Emissivity assumptions; urban rural; atmospheric corrections?

Please refer to point 1 above with regard to emissivity and atmospheric corrections.

(16) L237 - *reference*

We have added ref 47.

(17) L248 - *Relevance or implications of this period to the MODIS data*

This time mismatch is not a concern. Our interest is multi-year or climatic mean UHI, not its time trend or inter-annual variability. The modeled UHI is not sensitive to period chosen. In a follow-up study, we forced the model with a different atmospheric data for the period (2005-2015), and the UHI intensity was essentially the same as that obtained by using the Qian forcing data for 1972 to 2004 (Zhao et al., manuscript in preparation). The lack of sensitivity to climate period is also shown by Oleson (ref 37).

(18) L273 - *How defined?*

A definition is now given (Equation 3).

(19) L274 - *daytime/nighttime?*

The redistribution factor (f) was calculated separately for daytime and nighttime.

(20) L284 - *so the model is perfect? Comment on size of uncertainties.*

This is a good point. The reanalysis surface longwave radiation is biased low (ref 53). This has been taken into account in our analysis. We have improved the text for clarity:

“Because MERRA does not consider haze pollution but the observation was impacted by haze, some of the difference between the observed and reanalyzed L_{\downarrow} can be considered to result from haze pollution. In this study, the regression of the OMR L_{\downarrow} against the MODIS AOD is given as $[\text{OMR } L_{\downarrow}] = a + b \times [\text{AOD}]$, where the intercept a represents the reanalysis model bias⁵³ and the slope parameter b was taken as the sensitivity of L_{\downarrow} to AOD (Table 1).” (Lines 359-365)

(21) Figure 1 - Methods? Periods? What are data for?

Clarified.

(22) Figure 2 - What's the sign convention that is being used? Is storage - storage heat flux. Why is radiation not included? By convection - I assume turbulent sensible heat flux is meant (evaporation or latent heat flux is a turbulent or convective heat flux also). The anthropogenic heat flux should be large in Chinese cities. What are the source of the aerosols? If regional then U-R difference does not hold. If Urban then AHF large.

The sign convention is given by the new Equation 2. We have improved the figure caption. The parameterization of the anthropogenic heat flux is still primitive (Oleson ref 37), and is acknowledged in the text:

“One reason for the low bias is that the model scheme does not have a complete accounting of all sources of anthropogenic heat release in the urban land³⁷.” (Lines 215-217)

The question regarding the aerosol source is addressed in our response to review 3 (point 1).

Response to Reviewer 3

The paper "Urban heat islands in China enhanced by haze pollution" sets out to test the hypothesis that haze pollution, i.e., high concentrations of aerosols, in urban environments can contribute to the Urban Heat Island (UHI) effect by trapping some of the outgoing long-wave radiation in the urban domain. The authors claim to prove the existence of a significant impact of urban haze pollution (high aerosol concentrations) on the Urban Heat Island (UHI) effect by means of satellite data (MODIS). Furthermore, it aims to quantify the haze-pollution impact on UHI for a selected number of cities in China. Absorption of long-wave radiation depends to a large extent on the size of the particles which means that the retention of outward long-wave radiation in urban environments is due largely to the coarse fraction, i.e., dust particles and aged aerosol. Presumably, in coastal areas sea-salt aerosol could potentially contribute, too. The authors analyse satellite data to identify the haze-effect statistically in the observations. Quantification of the effect is attempted by applying a modelling approach in connection with re-analysis data and observations.

The paper presents novel research in the sense that it provides convincing evidence in support of a long-standing hypothesis. The work is novel also in that it extends existing research to a new domain - major cities in China - which due to the high level of aerosol pollution represent a promising research target for the considered effect. It is the relative dominance of haze or aerosol pollution in Asian cities, especially in China, that sets them apart from other major cities on the globe. In my opinion the paper is of moderate to significant interest to other researchers in the field. It presents convincing qualitative evidence for the existence of an impact of aerosol pollution on the UHI. That fact has important implications for pollution and urban climate change mitigation in that it outlines a co-benefit of reducing urban haze pollution.

In my opinion the claims made are convincingly underpinned by the evidence the authors have compiled in the paper, at least in a qualitative sense. The existence of a haze pollution impact on the UHI, at least during nighttime, is based on observational data and a simple but sound statistical analysis. The quantification also applies observations but does rely largely on a modelling approach and therefore has to be viewed, in my opinion, as a first estimate only which is exactly what the authors claim as well. Overall, I find the paper sufficiently convincing to justify publication in Nature Communications.

However, the main question that remains for me is in relation to the nature of the aerosol. The haze pollution effect on UHI is dominated by coarse aerosols, mainly dust I should think. It is vitally important to characterize and classify the sources of the coarse aerosol because the source determines the available mitigation options. Are the sources purely anthropogenic? If the main contributor is dust where does it originate? Is it road dust (e.g., unpaved road surfaces) or is coming from eroding soils outside the cities? If mitigation is a goal then these and further questions are essential. Maybe the authors could add a paragraph or two when they discuss the results.

The paper is presented in a clear fashion throughout. The abstract provides a concise and clear lead into the paper. With the exception of a few statements that could be improved and which I will discuss in detail under my specific comments the text is clear and easy to follow. In the introduction due attention is given to the existing literature with relevance to the subject and the background is discussed with sufficient detail. There is a clear thread throughout the section on the findings and the methodology is discussed in sufficient detail for the readers to understand and replicate the approach (provided they have access to the same data and model). I do not think that the paper should be shortened. In my opinion all the content is relevant and necessary. Figures and tables are very appropriate.

Therefore, I recommend for this paper to be accepted for publication in Nature Communications subject to a few revisions as outlined below.

Specific Comments:

(1) I have already mentioned that it would be good to include a very brief discussion on the composition of the aerosol that make up the urban haze since it is vitally important for mitigation strategies. One or two paragraphs at max should be sufficient.

This is an excellent suggestion. We have added the following text:

“In the semiarid climate zone in China, both the urban and its adjacent rural area are affected by coarse mineral particles transported from the Taklimakan Desert and the Gobi Desert³³. In the urban environment, road fugitive dust, construction-derived dust and domestic heating and cooking are additional sources of coarse mode aerosols, explaining why the ratio of PM₁₀ to PM_{2.5} concentration is greater in the semiarid cities in Northwest China than in the humid cities in South China³⁴⁻³⁶. The urban-rural AOD difference observed here appears to result from these urban anthropogenic sources.” (Lines 179-186)

(2) p3156-59: I think this statement should be changed to "can be more effective under specific circumstances". I find it a bit of an overstatement or at least a very broad brush to claim that aerosols in general are more effective in absorbing/emitting long-wave radiation than greenhouse gases or water vapour. Also it should be explained here that that the absorption effectiveness mostly

depends on aerosol size: large aerosols are effective LW-absorbers. What about aged aerosols? How does water vapour uptake by aerosols change that property and what does this imply for the different climatic zones that have been investigated.

Our point is that in the longwave atmospheric window, water vapor and greenhouse gases are not absorbing, but aerosols are. Nevertheless, to acknowledge the reviewer's concern, we have modified this portion of the text as:

"...But they are much more effective in absorbing and emitting radiation than water vapor and greenhouse gases in the longwave atmospheric window (wavelength range 8 to 12 μm) under specific conditions, thus having the potential to increase the overall longwave radiation energy received at the urban land surface¹¹. The overall effect depends on the initial particle size and size growth due to aging¹² and absorption of water vapor¹²." (Lines 57-60)

Note that in the revision, we have corrected the wavelength range for the longwave atmospheric window.

(3) p6l111-113: I suggest to order these factors in order of decreasing magnitude with respect to R2: 1.) P, 2.) delta-NDVI, and 3.) CF.

Done. (Lines 121-123)

(4) p6l130: Most likely L-down is the downward long-wave radiation fraction as a result of long-wave radiation being absorbed and re-emitted in the downward direction by the aerosols. However, it is never really defined as such. Can it be assumed the reader knows? A brief definition would help in my opinion, if nothing else then just to remove any doubt.

A definition is now given:

"Here L_{-} is the downward longwave radiation received by the surface including emission and scattering of air molecules and aerosols." (Lines 143-145)

(5) p7l158: A bit more detail on the statistics would help. Mean plus standard error/deviation?

Standard error is now included. (Line 174)

(6) p8l169-171: It wasn't immediately obvious to me what the offline calculation was and how it is used in the context of the paper. If I understand correctly, the offline calculation is used to quantify the radiative effect which cannot be determined from the model simulation. The total radiation effect is than a combination of model and offline calculation. Maybe I was just not paying enough attention but a sentence or two of clarification would help.

This portion of the text has been improved for clarity (Lines 192-197). In addition, we have expanded the description of the online and offline UHI calculations (Lines 333-337).

(7) p12l277-280: Could you add a discussion of other potential differences that may exist between observations and the reanalysis data? How firmly is the assumption justified that the difference between model and reanalysis is ONLY due to aerosols? Surely, the model behind the reanalysis data has a different physical core than the urban model that is applied. There are other significant differences. A brief discussion of these differences and their potential impact on the findings would help I believe.

Please refer to our response to review 2, point 20.

(8) p15/351: *Is there maybe another reference for this? I am not sure Discussion papers are acceptable references any longer.*

This paper is now published (ref 28).

Reviewer Comments

Reviewer #1 (Remarks to the Author):

In the revised MS, the authors have sufficiently addressed my concerns. I would recommend it being accepted for publication.

Reviewer #2 (Remarks to the Author):

There are some key methods that remain unexplained.

L145 - English

156 - basis for lambda 0 value selected?

L171 - need to be more specific

L174 - english

L175/176/177 - less efficient energy redistribution? Is this regional source or something about the industry in these areas? - partially discussed in the next paragraph

L198-200: - still need to indicate this is expected e.g. Oke review of urban climate chapter in the book- Climate of Canada

L209 - and observed much earlier than this

L260: - given that the extent of a large city - 0.1 deg latitude is small relative to the potential central location - doesn't this make this it a challenge?

L267/273 - given the issues with urban emissivities (with work on going with respect to this) it should be documented what the size of this impact could be

L298 - do negative differences occur? Are only positive differences being included in the mean?

L324 - subscript f1 and f2 are not defined

Term1 and term 2 are both convective - rephrase

Term 1(l329) includes long wave or has this been removed - if so change notation

L346 - explain how the Bowen ratio is determined.

L358/359 - indicate if these are urban or rural sites.

Fig1 - need to indicate which point is which city (e.g. number/letter all and give a list of names in the supplemental materials that correspond to the number). Need to be clear what data (time period etc) are in each point

Figure 2 - colour does (and hence caption) does not hold up in B& W

Convection and evaporation - see comment above

Reviewer #3 (Remarks to the Author):

This is a review of the revised version. I have studied the revisions in detail and I am convinced that my previous concerns with the manuscript have been addressed adequately and in full. It also seems

that the other two reviewer's concerns have been addressed adequately but this is not really for me to decide.

Therefore, I am recommending this manuscript to be published as is.

Response to Referees

Response to reviewer #2

Reviewer #2 (Remarks to the Author):

There are some key methods that remain unexplained.

(1) L145 - English

English checked.

(2)156 - basis for lambda 0 value selected?

This parameter is now defined as $\lambda_0 = 1/(4\sigma T_s^3)$

(3) L171 - need to be more specific

We explain that the reference plane is at the first model grid height.

(4) L174 – English

Checked.

(5) L175/176/177 - less efficient energy redistribution? Is this regional source or something about the industry in these areas? - partially discussed in the next paragraph

We have added a short explanation:

“...less efficient energy redistribution (smaller f values due to larger aerodynamic resistance and Bowen ratio, Equation 5) ...”

(6) L198-200: - still need to indicate this is expected e.g. Oke review of urban climate chapter in the book- Climate of Canada

We have added this chapter as ref 38. The sentence has been rewritten as “This offsetting effect of aerosols on radiation has been reported previously by ref ³⁷ and the year-long observations at an urban-rural site pair support this interpretation²⁹”

(7) L209 - and observed much earlier than this

We are not aware of any observational studies prior to the publication referenced here (ref 4) showing that changes in convection efficiency is a major contributor to the UHI.

(8) L260: - given that the extent of a large city - 0.1 deg latitude is small relative to the potential central location - doesn't this make this it a challenge?

The sentence has been improved for clarity:

“The selected urban-rural pixel pairs do not differ by, on average, more than 100 m in elevation and by more than 0.1° in latitude.”

(9) L267/273 - given the issues with urban emissivities (with work on going with respect to this) it should be documented what the size of this impact could be

We have added a sentence here:

“The impact of emissivity on the surface temperature retrieval has been discussed elsewhere^{43,46}.”

(10) L298 - do negative differences occur? Are only positive differences being included in the mean?

We used both negative and positive AOD differences to compute the annual mean value.

(11) L324 - subscript f_1 and f_2 are not defined

Term 1 and term 2 are both convective - rephrase

Term 1(L329) includes long wave or has this been removed - if so change notation

We have added the definition of Δf_1 and Δf_2 .

ΔR_n , the net apparent radiation difference between urban and rural area, includes longwave radiation, so term 1 has been rephrased.

(12) L346 - explain how the Bowen ratio is determined.

Done.

(13) L358/359 - indicate if these are urban or rural sites.

We have added “Yongfeng (Jiangsu Province) and Luancheng (Hebei Province) are cropland sites and Xilinguole is a grassland site in Inner Mongolia.”

(14) Fig1 - need to indicate which point is which city (e.g. number/letter all and give a list of names in the supplemental materials that correspond to the number). Need to be clear what data (time period etc) are in each point

All the data are from 2003 to 2013 and this was written as ‘**Figure 1: Nighttime MODIS surface urban heat island intensity in mainland China from 2003 to 2013.**’ We have numbered all the cities (Figure 1a). The city names and coordinate information have been list in supplementary table 1.

*(15) Figure 2 - colour does (and hence caption) does not hold up in B& W
Convection and evaporation - see comment above*

We have changed the convection term to sensible heat in the legend.